# CFD Investigation of Trout-Like Configuration Holding Station near an Obstruction

**Kamran Fouladi** [1,*] **and David J. Coughlin** [2]

1. Department of Mechanical Engineering, Widener University, Chester, PA 19013, USA
2. Department of Biology, Widener University, Chester, PA 19013, USA; djcoughlin@widener.edu
* Correspondence: kfouladi@widener.edu; Tel.: +1-610-499-1292

**Abstract:** This report presents the development of a fluid-structure interaction model using commercial Computational fluid dynamics software and in-house developed User Defined Function to simulate the motion of a trout Department of Mechanical Engineering, Widener University holding station in a moving water stream. The oscillation model used in this study is based on the observations of trout swimming in a respirometry tank in a laboratory experiment. The numerical simulations showed results that are consistent with laboratory observations of a trout holding station in the tank without obstruction and trout entrained to the side of the cylindrical obstruction. This paper will be helpful in the development of numerical models for the hydrodynamic analysis of bioinspired unmanned underwater vehicle systems.

**Keywords:** swimming kinematics; biomimetic; fish motion; unmanned; UUV; CFD; FSI; UDF





## 1. Introduction

Unmanned underwater vehicle (UUV) systems play a crucial role in remote Intelligence, Surveillance, and Reconnaissance (ISR) missions in marine environments. These systems are touted or have already been deployed for various critical missions, such as port protection surveillance, deep ocean exploration, seabed surveying, mine countermeasures, and underwater structure inspection [1–4]. A comprehensive historical overview of advances and development of UUV systems can be found in a report by Budiyono [5].

Many earlier UUVs have been designed with rigid bodies and driven using propellers [6]. Additionally, these systems use rigid control surfaces for maneuvering. In the past decade, the design and development of the UUV configurations have progressed significantly, with notable advances achieved in propulsion efficiency and maneuverability capabilities. The new breed of UUVs have been inspired by natural swimmers such as fish. Fishes are remarkable in their ability to control and maneuver using flexible bodies and fins [6]. They achieve high propulsive efficiency and excellent maneuverability through the coordinated motion of the body, fins, and tail [7].

Many of the more recent UUVs benefit from the biomimetic propulsion system as an alternate to powered thrusters. Several current efforts have been inspired by biological mechanisms to design fish robots that have achieved higher propulsion efficiency and the capability to maneuver [8–12]. These improvements are generally made through the coordinated motion of various body parts such as body, fins, and tail. Furthermore, the bioinspired undulating mechanisms have received significant attention from engineers and biologists, since many natural swimmers incorporate body part undulation in their motion [13]. Some efforts have incorporated undulating mechanisms in the design of UUV systems, and recent examples are works by Park et al. [14], Chowdhury et al. [15], Liu et al. [16], and Park et al. [17].

There are also efforts to design bioinspired UUVs to operate in tandem or in groups to increase their propulsion efficiency, with each UUV taking advantage of the lower pressure field in the wake of a neighboring UUV. Recent numerical simulations have shown the

benefits of fish positioning themselves strategically in the wake of other swimmers to intercept their shed vortices and increase their propulsive efficiency [18]. Therefore, the group and tandem UUV design efforts can also benefit from findings in research works that have documented the energetic benefits of individual fish swimming near an obstruction and turbulence [19–22]. Mainly, obstacles in the moving water create flow conditions and vortices that an individual fish can benefit from to reduce its energetic cost of a holding station in the water. By adopting a distinctive swimming pattern, the fish body can act as a self-correcting hydrofoil resulting in reduced muscle activity during vortex exploitation [23].

A bioinspired UUV must be capable of flapping motion with the deformable body, fins, or tails. The interactions between various components of a bioinspired UUV require an accurate and coupled fluid-structure interaction (FSI) analysis [24]. Therefore, there has been a notable uptake in the use of the FSI approach in these studies [15,25–28]. Moreover, many researchers have simulated the three-dimensional undulation motion of either natural swimmers or UUVs with the flexible body utilizing the FSI approach with the use of User Defined Functions (UDF) in a CFD simulation.

Guan et al. presented a 3D CFD simulation of a biomimetic robot fish [29]. They modeled the movement of their robotic fish using a UDF based on a carangiform propulsion model where the wavelike motion takes place in the rear half of the fish. Li et al. investigated using the hydrodynamics of a 3D tuna-like body performing thunniform bioinspired swimming [30]. Their hydrodynamic analysis on different parts of the fish body showed that the caudal fin is a significant source of thrust production. In contrast, the deforming rear body was shown to have minimal effects on the thrust force.

In their numerical simulation study, Xuel et al. analyzed three types of motion models attributed to tuna, which will be helpful for the control of a bioinspired UUV [31]. The CFD analysis carried out by Patil et al. investigated the hydrodynamic performance of the carangiform motion of a fish body similar to tilapia [32]. Their study consisted of a fish body moving in a plane with a periodic oscillation of tail and abdomen in an axis perpendicular to the plane.

*Objective and Motivation*

The review of previous studies indicates that the energetic implications of group locomotion in schools of fish are not fully understood. Therefore, further investigation on exploring the propulsive benefits or disadvantages of swimming in schools is warranted and can help in the design of UUV systems. Our research effort is aimed at taking a step in that direction by contributing to the understanding of fish interacting with vortices generated by obstructions. The present paper reports on the development of an accurate and efficient fluid-structure model using the commercial software ANSYS Fluid® (Fluent®, Canonsburg, PA, USA) [33] and an in-house UDF to model locomotion of a fish. This paper reports on our attempt to verify this simulation model using the observations reported by Cook and Coughlin [20]. Hence, the model setup of the present study will closely mimic the experimental setup presented in Refrence [20], in which they examined the effect of a vertical cylindrical obstruction placed in a steady flow on swimming by rainbow trout *Oncorhynchus mykiss* in a respirometry swim tunnel. To accomplish the above objective, the configuration used in the numerical simulations will be geometrically similar to the trout in Reference [20] placed in similar-sized rectangular tank. The model geometry used in the present study is dubbed a trout-like configuration and will be presented as TLC in the remainder of this report.

## 2. Materials and Methods

The flow physics in the vicinity of the oscillating geometries are complex and vastly different than the flow about rigid bodies. These flows are expected to include separations, recirculations, large vortices, and boundary layers with strong adverse pressure gradients. The present investigation employed an FSI approach using ANSYS Fluent®, along with

UDF and dynamic meshing. The in-house-developed UDF model was employed to describe the movement of the trout, and dynamic meshing was used to accommodate the moving geometry and reduce the mesh distortion. The UDF used in the present approach is presented in Appendix A.

### 2.1. CFD Tool

The finite volume-based commercial software ANSYS Fluid$^{®}$ was employed to solve the transient incompressible continuity and momentum governing equations, as shown below.

$$\frac{\partial \rho}{\partial t} + \frac{\partial}{\partial x_i}(\rho u_i) = 0 \tag{1}$$

$$\frac{\partial}{\partial t}(\rho u_i) + \frac{\partial}{\partial x_j}(\rho u_i u_j) = -\frac{\partial P}{\partial x_i} + \frac{\partial \tau_{ij}}{\partial x_j} \tag{2}$$

where $u_i$ is the velocity component in Cartesian direction $i$, and $\tau_{ij}$ is the stress tensor arising from shear forces, which can be expressed in terms of velocity field and fluid viscosity. Additionally, $P$, $\rho$, and $t$ are the static pressure, density, and time, respectively. A pressure-based solver with a "coupled" pressure–velocity coupling scheme was used with second-order spatial discretization for the pressure and second-order upwind scheme for the momentum equation. The time-averaged drag coefficient $C_{D\_Mean}$ is defined as

$$C_{D\_Mean} = \frac{\overline{F}_x}{0.5\rho A(U)^2} \tag{3}$$

where $\overline{F}_x$ is the time-averaged axial force, $U$ is the freestream velocity, and $A$ is the surface area.

Using water as the working fluid with constant properties at T = 10 °C, the velocity inlet (0.768 m/s) and pressure outlet (ambient) were used for the inlet and outlet boundaries, respectively. All other boundaries and surfaces were declared as no-slip walls. The Reynolds number in the present study, estimated approximately at $1.97 \times 10^5$, is defined as

$$Re = (\rho\, U\, L)/\mu \tag{4}$$

where $U$ (0.768 m/s) and $L$ (0.256 m) are the average tank velocity and TLC length, respectively.

The two-equation realizable $k - \varepsilon$ turbulent model was used in the present study to capture the turbulence effects. Rahman et al. argued that the realizable $k - \varepsilon$ model is an accurate and efficient model for flows about oscillating geometries at low Reynolds numbers, less than $Re_{critical} = 500,000$, with complex flow phenomena such as boundary layer separations and vortical wakes [34].

In the present approach, body deformations resulting from the fluid loads such as pressure and viscous forces were ignored. However, the fluid–structure interaction was used to account for the TLC shape changes, as the UDF was used to set the tail-segment of the TLC in a flapping motion. For these cases, a dynamic mesh or a deforming mesh method is attractive to ensure accuracy, as the cells in the computational mesh must move to follow the changing shape of the boundary. The dynamic meshing technique generates a new mesh at corresponding to the changing geometrical shapes. More specifically, in this study, the dynamic mesh allowed for the displacement of the domain boundaries in the simulation, forming new mesh according to the geometrical changes as the TLC was set in motion. Due to the importance of the dynamic mesh performance to accurately adapt the mesh to each TLC posture, both the smoothing and remeshing mesh methods were used in the present approach.

### 2.2. Model Description

Cook and Coughlin [20] examined the interaction of an individual trout with a cylindrical obstruction in a steady flow in a respirometry swim tunnel. In their setup, individual trout, approximately 0.256 *m* long, were swum in a swim tunnel with a 0.875 m × 0.25 m × 0.25 m test section. To verify our simulation model through a comparison against the findings reported by Cook and Coughlin, the present study employed a computational model with a setup similar to the experimental setup used in Reference [20]. The computational TLC model was similar, geometrically and dimensionally, to the trout used in Reference [20]. Additionally, the obstruction and tank had the same dimensions in both the simulation and the experiment setups. The schematic of the computational domain with TLC in the vicinity of obstruction is presented in Figure 1. The obstruction is a 6-cm diameter cylinder placed vertically in the center of the test chamber. It should be noted that Cook and Coughlin examined the effects of the obstruction at three different stream velocity magnitudes of 0.256 m/s, 0.512 m/s, and 0.768 m/s. However, the highest velocity (0.768 m/s) was only considered in the present study.

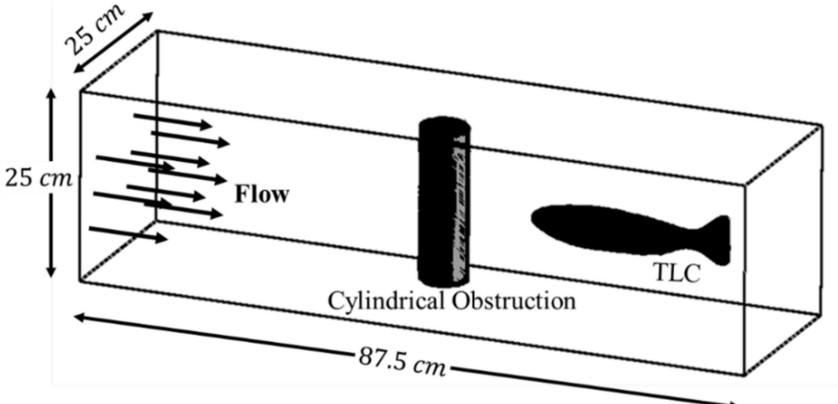

**Figure 1.** Computational schematic of the TLC downstream of a cylindrical obstruction.

A grid independence study was conducted to determine the most efficient mesh with reasonable simulation time. The grid independence study included a total of five meshes with different grid densities for TLC alone oscillating at 2.5 Hz. The five trials had target mesh sizes ranging from 150,000 to 600,000 cells. Figure 2 presents the results of the grid independence study, and it indicates a consistent value for the time-averaged drag coefficient, $C_{D\_Mean}$, values for all the meshes. Based on the grid independence study, the mesh with 210,000 cells was chosen due to its lower run time and no significant degree of error. The mesh used for this study was generated to adequately resolve the wake details, as well as the viscous sublayer with $y^+ \approx O(1)$. The time step used for these simulations ranged from 0.0005 s to 0.001 s, with 50 maximum iterations per time step. A representative of the mesh at the plane of symmetry is shown in Figure 3.

### 2.3. Oscillatory Motion of the TLC

The trout are sub-carangiform swimmers. Berder described that the motion by this group has a noticeable increase in wave amplitude along the body, while the majority of the undulation work is done by the rear half of the fish [35]. Based on the analysis of the video footage from Cook and Coughlin's study [20], it was determined that body undulation of the trout in their study takes place in the last 60% of the body, labeled the tail segment here. However, the first 40% of their bodies, dubbed the head segment here, remain almost unchanged. It was also determined that a spline curve best represents the centerline shape of the configuration at a given instant during the undulation. Furthermore, the undulation takes the form of a sinusoidal wave with a specific wavelength and frequency in the tail segment, with the amplitude reaching the maximum at the tail end. The amplitude envelopes, along with the motion trajectories of the TLC centerline at various times during

the undulation, are presented in Figure 4. The amplitude envelope describes the maximum swing state of the centerline curve.

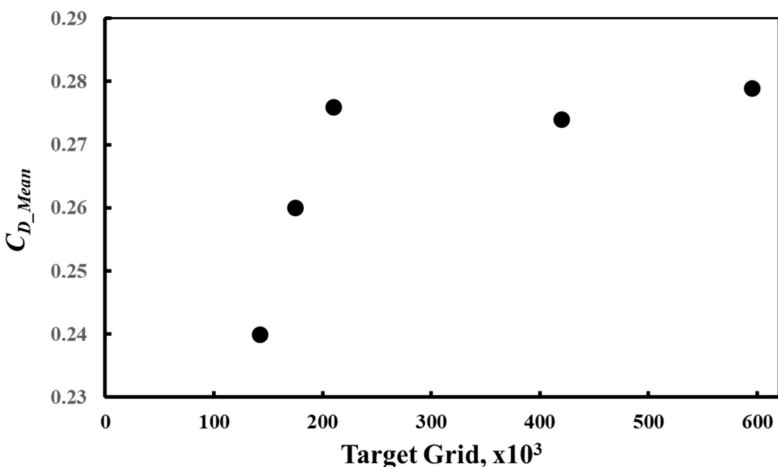

**Figure 2.** $C_{D\_Mean}$ for the grid independency study.

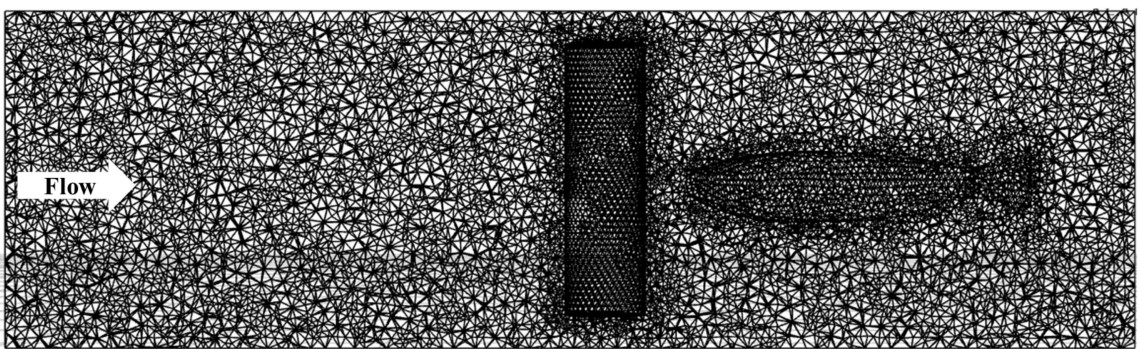

**Figure 3.** Mesh distribution on the plane of symmetry (side view).

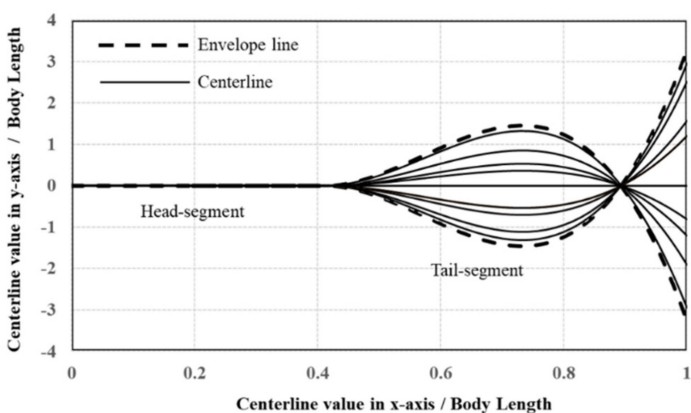

**Figure 4.** Motion trajectories of the centerline of the TLC.

Equation (5) describes the transverse undulation of the centerline coordinates of the tail segment of the TLC as

$$y(x,t) = A(x) \sin(2\pi f t - kx) \tag{5}$$

where $f$ is the motion frequency, $t$ is the time, $k$ is the wavenumber, and $A(x)$ is the amplitude envelope. The wavenumber $k$ is set at zero in the present simulation to simulate

the observations presented by Cook and Coughlin [20]. The amplitude envelope for sub-carangiform swimmers such as rainbow trout is also approximated by a second-order polynomial [36]:

$$A(x) = C_1 x + C_2 x^2 \qquad (6)$$

where $C_1$ is the linear wave amplitude envelope, and $C_2$ is the quadratic wave amplitude envelope. These coefficients are related to the size of the body, speed of swimming, posture, and other factors.

In the present study, the user-defined function DEFINE_GRID_MOTION UDF was used to set the tail segment of the TLC into a flapping motion, according to Equation (5). This UDF allows for updating the position of each node based on the deflection due to undulations. Subsequently, the volume grid was regenerated and smoothed during the motion at each iteration within each time step by the remeshing and smoothing approaches in ANSYS Fluent®.

## 3. Results and Discussion

The computational model developed in this study was used to simulate the physical testing of trout swimming near an obstruction in a swim tunnel conducted by Cook and Coughlin [20]. The computational model was based on a similar geometrical and flow conditions to the experimental setup. As previously mentioned, three different flow velocities were considered in Cook and Coughlin's experimental study. However, only the simulations based on the highest velocity magnitude (0.768 m/s) are presented here.

Cook and Coughlin presented three cases in their assessment of the obstruction effect on the energy consumed by the trout. The first case represented the fish holding station in the flowing water without any obstruction present. With a vertical obstruction present, the fish was observed to entrain either to the side or behind the obstruction in the second and third cases, respectively. The CFD simulations of the TLC in the above scenarios are presented in the following sections.

### 3.1. TLC Alone (No Obstruction)

In the first phase of the study, simulations were conducted to compare with observations from the experimental efforts with the trout holding station without the presence of an obstruction in a moving stream. With a freestream velocity of three body lengths per second (0.768 m/s), Cook and Coughlin observed a fish holding station with the tail-beat frequency of about 4 Hz. The TLC simulations with no obstruction in the tank were conducted at various oscillating frequencies (2.5 Hz, 4 Hz, 6 Hz, and 8 Hz). An example of the oscillation effects on the flow field is presented in Figure 5, where the variation of instantaneous streamlines on the tank mid-plane during an oscillation cycle are shown for the 2.5-Hz simulation case.

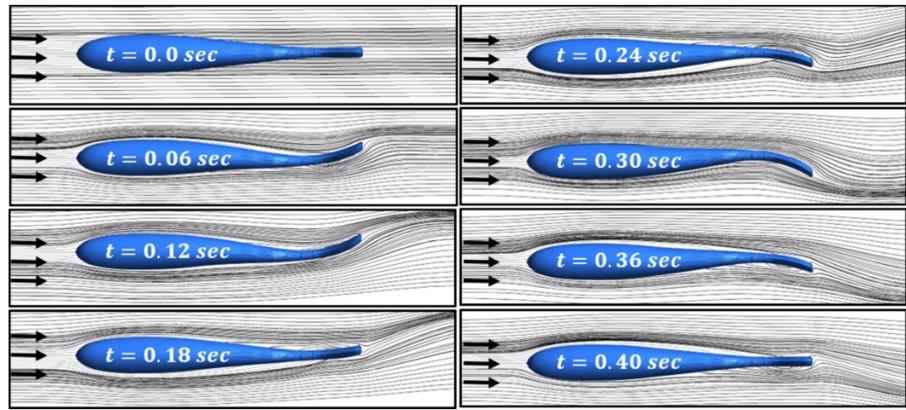

**Figure 5.** Instantaneous streamlines on the tank mid-plane (2.5 Hz).

Additionally, the results of these simulations are compared against the experimental observations and are used to investigate the effects of tailbeat frequency on the drag forces experienced by the geometry. The results of this comparison are presented in Figures 6 and 7. The time variation of the drag coefficients in Figure 6 show consistently higher peaks at a 2.5-Hz frequency compared to the other tailbeat frequencies. This variation results in a significantly larger time-averaged drag coefficient at 2.5 Hz compared to values at the higher frequencies. However, the time-averaged drag coefficient, $C_{D\_Mean}$, values are predicted to be similar for 4 Hz and 6 Hz (Figure 7), and they are much lower than the magnitude of $C_{D\_Mean}$ for 2.5 Hz. It should be noted that trout employ slow-twitch aerobic muscle for station holding swimming, such as modeled here, which makes higher frequencies (e.g., 6 Hz and 8 Hz) not sustainable from a muscle physiology standpoint [37]. Therefore, it can be deemed that 4 Hz would be the most efficient oscillation frequency for TLC, overcoming approximately the same drag force with a lower amount of oscillation energy. These results are consistent with the observations made in Cook and Coughlin's study, where they reported that the trout were seen to hold station by oscillating at 4 Hz.

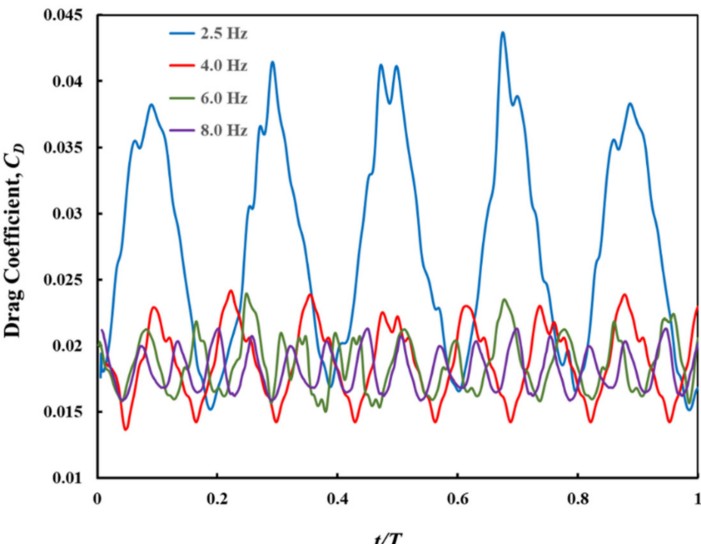

**Figure 6.** Instantaneous drag coefficient for the station TLC with no obstruction.

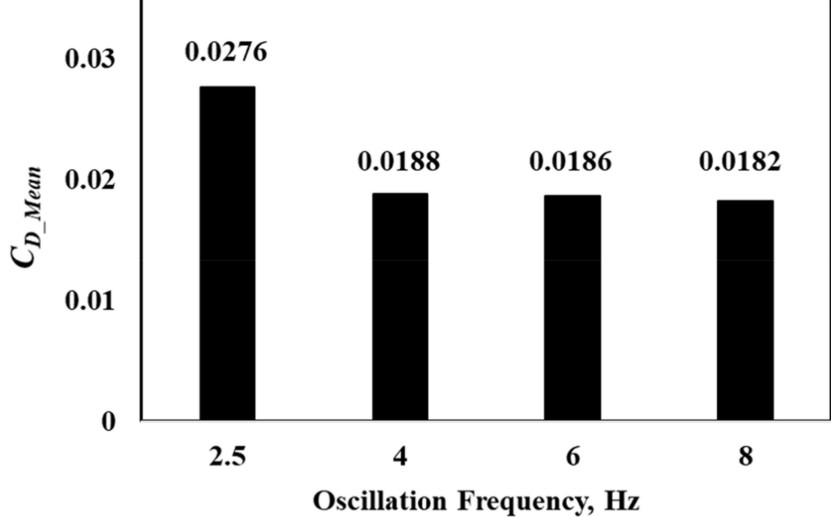

**Figure 7.** Variation of $C_{D\_Mean}$ with the oscillation frequency with no obstruction.

### 3.2. TLC Positioned Lateral to the Obstruction

Cook and Coughlin [20] observed that, when the obstruction was present, the fish spent the majority of the test period (estimated in excess of 90% of the test period) entrained in a position lateral to the obstruction, with the tip of their head at approximately the mid-point of the cylinder. The development of Venturi effects due to the presence of the obstruction in the tank (reduced cross-sectional area) can help explain this behavior. The Venturi effect refers to a reduction in the pressure field and an increase in the fluid velocity, which is due to the smaller cross-sectional area. Figure 8 displays the pressure field without the TLC in the tank, indicating the lowest flow pressure aligned with the obstruction centerline. The pressure distribution changes with TLC at the side of the obstruction, as depicted in Figure 9. Superimposed on this figure is the time-averaged drag coefficient for nonoscillatory TLC, with the tips of their heads positioned at various locations (designated with markers) relative to the obstruction. The drag coefficient comparison shows that TLC would experience the lowest drag if the tip of the head was aligned with the mid-point of the cylinder. Therefore, the next set of simulations were conducted with TLC oscillating at various tailbeat frequencies and placed to the sides of the cylinder aligned with its mid-point, as observed in Reference [20].

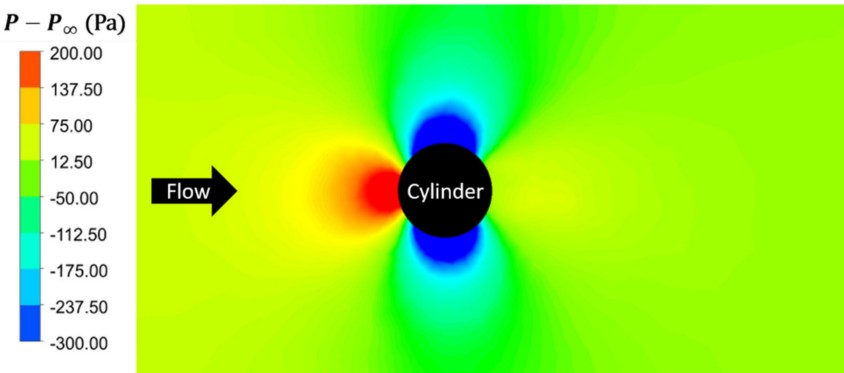

**Figure 8.** Effects of obstruction (no TLC) on the pressure field (top view).

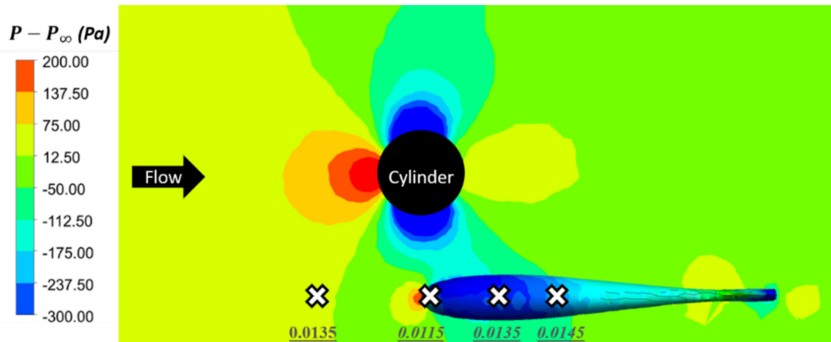

**Figure 9.** Variation of $C_{D\_Mean}$ with the position (relative to obstruction) for nonoscillatory TLC.

The comparisons of the time variations of the drag coefficients at various tailbeat frequencies for TLC entrained to the side of the obstruction are presented in Figure 10. The comparisons show significantly higher peaks for the tailbeat frequency of 1 Hz, which is even higher than no oscillation. That is, the lower tailbeat frequency could have a negative impact on the drag coefficient. However, the drag performance improves at higher tailbeat frequencies, with the lowest value recorded at 2.5 Hz (Figure 11), as observed by Cook and Coughlin.

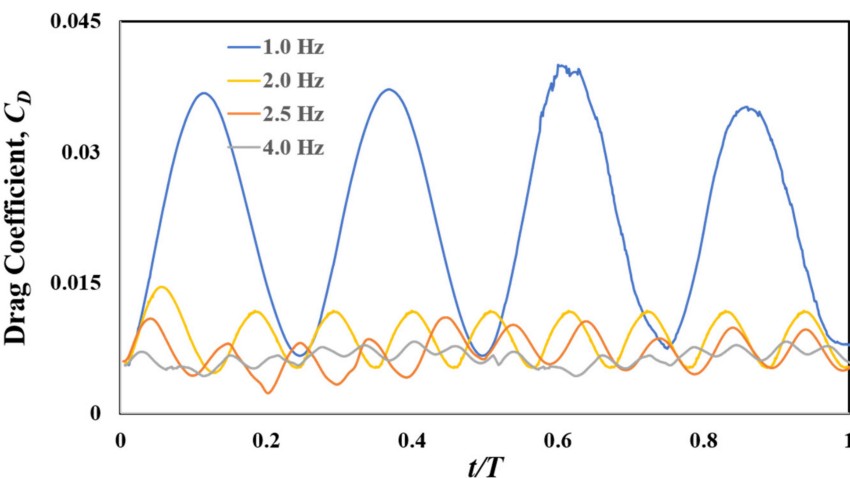

**Figure 10.** Instantaneous drag coefficient for station TLC entrained to the side of the obstruction.

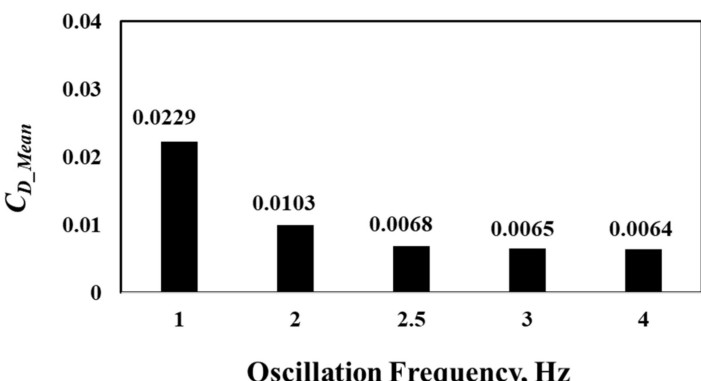

**Figure 11.** $C_{D\_Mean}$ vs. oscillation frequency for TLC entrained to the side of the obstruction.

### 3.3. TLC Positioned Downstream of the Obstruction

It has been shown that fish may take a position behind an obstacle in moving water to take advantage of the vortices generated by the obstacle to reduce its energetic cost of holding station in water. The oscillatory motion used in the previous cases presented earlier was also used to investigate the effect of an obstruction when the TLC is positioned downstream of the obstruction. Figure 12 shows the interaction of TLC with the obstruction's wake vorticity distribution.

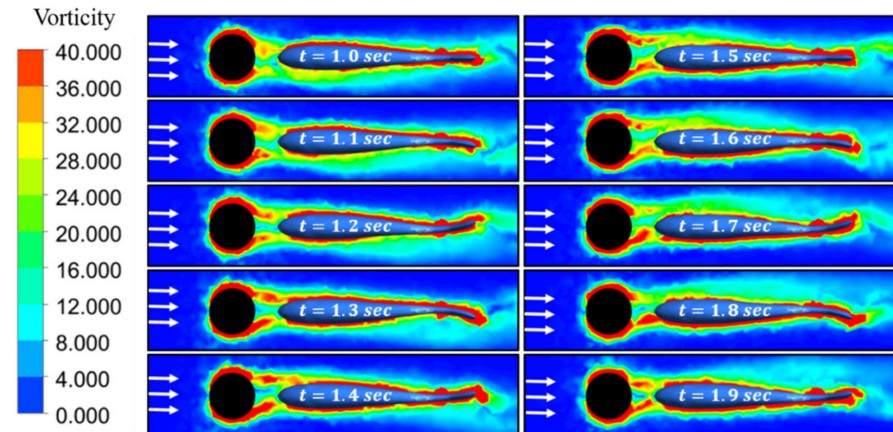

**Figure 12.** Snapshots of the vorticity field with TLC oscillating at 4 Hz behind the cylinder.

Cook and Coughlin [20] observed that trout entrained behind the obstruction held station at length equal to obstruction's diameter, with a tailbeat frequency of about 2.5 Hz. The simulation parametric study also showed the lowest drag forces when the TLC was positioned about a diameter away. However, the results did not confirm the lowest occurring at 2.5 Hz when the TLC was placed at that distance behind the obstruction. Figures 13 and 14 present the instantaneous drag coefficients and time-averaged coefficients for TLC oscillating at various tailbeat frequencies.

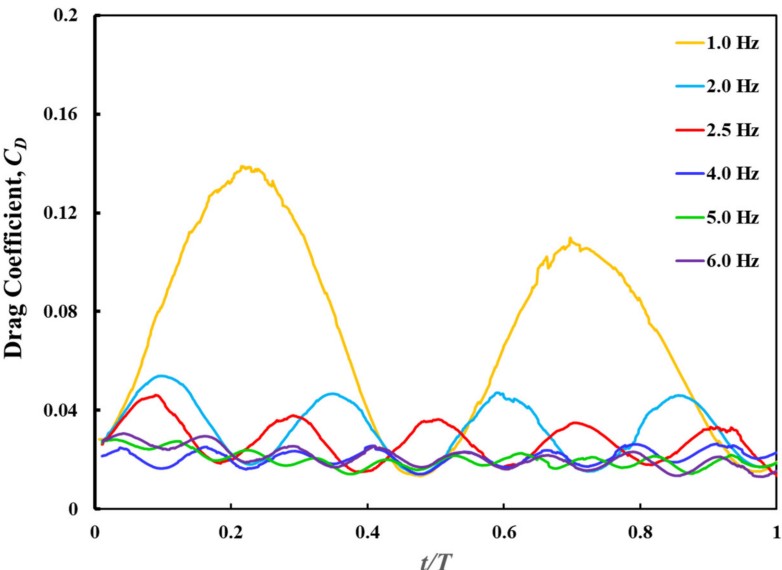

**Figure 13.** Instantaneous drag coefficient for the station TLC entrained downstream of the obstruction.

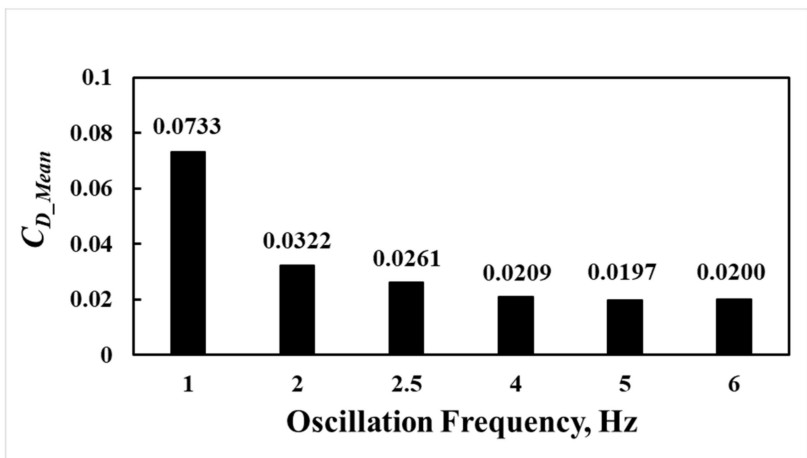

**Figure 14.** $C_{D_{Mean}}$ vs. oscillation frequency for TLC entrained downstream of the obstruction.

The discrepancy between the laboratory observations and simulation results may be due to the differences between the oscillatory motions of the trout and TLC. Previous laboratory studies have reported that a trout holding station behind the cylinders adopts a distinctive swimming pattern, which has been termed the Kármán gait [20,38]. Liao et al. described that the body amplitudes and curvatures of the trout during a Kármán gait are much larger than those of trout swimming at an equivalent flow velocity in the absence of a cylinder. According to Akanyeti and Liao [39], Kármán gaiting fish exhibit substantial lateral translations and body rotations, and these motions are periodic, with frequencies that match the frequency of vortex shedding. Akanyeti and Liao stated that there was an inverse correlation between the head angle and body angle. They presented a mathematical model that described how fish swim in vortical flows in real time. Therefore, the oscillation

motion for TLC downstream of an obstruction in future simulations needs to be adjusted to the Kármán gaiting pattern and should be based on the model proposed by Ankayaeti and Liao [39].

## 4. Conclusions

This paper reported on the use of a CFD model to simulate and understand the motion of a sub-carangiform swimmer holding station in a moving water stream with and without an obstruction. The model assumed undulation at the tail segment (last 60%) of the body in the form of a sinusoidal wave with a specific wavelength and frequency, with the first 40% remaining almost unchanged. The oscillation model used in the simulation was based on observations of individual trout swimming in a respirometry tank in a laboratory experiment, with no obstruction in the tank. The simulations of locomotion of a trout-like configuration (TLC) and its complex interactions with the flow field of a vertical obstruction were achieved using an FSI method with an in-house-developed UDF.

The results generated from the simulation of TLC alone in the tank (no obstruction) and TLC entrained to the side of the obstruction were consistent with the observations made in the laboratory experiment. Additionally, predicted for the side entrainment case was the Venturi effects generated due to the presence of the obstruction in the tank, which may explain the reason trout preferred to position themselves to the side of the obstruction in the laboratory experiment. However, the simulation findings are inconclusive for the case of TLC entrained downstream of an obstruction. These results may indicate an inconsistency between the TLC oscillatory motion and the motion by trout in the laboratory experiment.

The first phase of this research effort was focused on a natural swimmer taking advantage of an obstruction in the flow path. However, the numerical model employed in this study can be enhanced with data obtained from additional experimental works using mechanical fish. Therefore, our future work will include experimental efforts using mechanical fish configurations capable of mimicking various natural swimmers' oscillatory motions in a laboratory setting. Furthermore, a thermodynamic second law analysis will be incorporated into the simulation model for determining the optimum UUV performance in various scenarios.

**Author Contributions:** Conceptualization, K.F. and D.J.C.; methodology, K.F.; software, K.F.; validation K.F. and D.J.C.; formal analysis, K.F. and D.J.C.; investigation, K.F.; resources, K.F.; data curation, K.F.; writing—original draft preparation, K.F.; writing—review and editing, D.J.C.; visualization, K.F.; supervision, K.F.; and project administration, K.F. All authors have read and agreed to the published version of the manuscript.

**Funding:** This research received no external funding.

**Institutional Review Board Statement:** Not applicable.

**Informed Consent Statement:** Not applicable.

**Data Availability Statement:** Not applicable.

**Conflicts of Interest:** The authors declare no conflict of interest.

## Appendix A

TLC oscillatory motion UDF script:

```
1:    /*******************************************
2:    node motion based on simple fish undulation equation
3:    compiled UDF
4:    *************************************************** /
5:    #include "udf.h"
6:    DEFINE_GRID_MOTION(trout, domain, dt, time, dtime)
7:    {
8:        Thread *tf = DT_THREAD(dt);
```

```
9:      face_t f;
10:     Node *v;
11:     real NV_VEC(omega), NV_VEC(axis), NV_VEC(dx);
12:     real NV_VEC(origin), NV_VEC(rvec);
13:     real sign;
14:     real ap1;
15:     real ap2;
16:     real ap3;
17:     real xgeom;
18:     real segment;
19:     real xr;
20:     real yn;
21:     real a;
22:     real b;
23:     real c;
24:     real d;
25:     int n;
26:     /* set cubic spline paramters*/
27:     a = 0;
28:     b = 1;
29:     c = -174.98;
30:     d = 1304.43;
31:     /* set deforming flag on adjacent cell zone */
32:     SET_DEFORMING_THREAD_FLAG(THREAD_T0(tf));
33:     segment = 0.4;
34:     xgeom = 0.26;
35:     xr = xgeom*segment;
36:        Message("segement = %f, xgeom = %f\n", segment, xgeom);
37:     Message("xr = %f, d = %f\n", xr, d);
38:     sign = cos(15.7 * time);
39:     Message("time = %f, omega = %f\n", time, sign);
40:     NV_S(omega, =, 0.0);
41:     NV_D(axis, =, 0.0, 1.0, 0.0);
42:     NV_D(origin, =, 0.0, 0.0, 0.0);
43:     begin_f_loop(f, tf)
44:     {
45:        f_node_loop(f, tf, n)
46:        {
47:          v = F_NODE(f, tf, n);
48:          /* update node if x position is greater than 0.02
49:          and that the current node has not been previously
50:          visited when looping through previous faces */
51:          if (NODE_X(v) > xr && NODE_POS_NEED_UPDATE(v))
52:          {
53:             /* indicate that node position has been update
54:             so that it's not updated more than once */
55:             NODE_POS_UPDATED(v);
56:             ap1 = NODE_X(v) − xr;
57:             ap2 = a + (b * ap1) + (c *ap1 *ap1) + (d *ap1*ap1*ap1);
58:             /* dx[1] = sign * ap2;
59:             /* dx[0] = 0.0;
60:             /* dx[2] = 0.0;
61:             /* NODE_COORD(v)[1] = yn; */
62:             omega[1] = sign * pow(ap2 / 1.0, 1.0);
```

```
63:          NV_VV(rvec, =, NODE_COORD(v), -, origin);
64:          NV_CROSS(dx, omega, rvec);
65:          NV_S(dx, *=, dtime);
66:          dx[0] = 0.0;
67:          NV_V(NODE_COORD(v), +=, dx);
68:        }
69:      }
70:    }
71:    end_f_loop(f, tf);
72: }
```

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
