# Peer review of "CFD Investigation of Trout-Like Configuration Holding Station near an Obstruction"

_fluids, doi:10.3390/fluids6060204_

Round 1
Reviewer 1 Report
1) Please present a mesh convergence study to show that a) the results are independent of the mesh used; b) the mesh used is optimal;
2) It would be useful for the readers if the authors can append the UDF code in the appendix
3) Some quantitative comparison with experimental observations/measurements would help to improve reliability of the results presented
Reviewer 2 Report
Overview:
This study presents computational simulations of fluid dynamics in the motion of a sub-carangiform swimmer holding station in a moving water stream. ANSYS Fluent is used for modeling the fluid structure interaction and an in-house UDF was developed to describe the movement of the trout. The paper uses a previous experimental work by Cook and Coughlin (2010) to problem definition and draw meaningful comparisons. Different steps of the simulation include 1) isolated trout-like configuration (TLC), 2) TLC positioned lateral to the obstruction and 3) TLC positioned downstream of the obstruction. The first two parts of the simulation agree well with experimental observations but inconsistent behavior was observed for the third part. This lack of agreement was explained in the light of the differences between the TLC oscillatory motion and trout motion in the laboratory settings.
Overall Evaluation:
This study appears technically valid and has a sound organization in presenting materials and the model, providing brief description of the computational approach, and reporting results and of each step
and the discussion. Additional comments regarding Venturi effects for the side entrainment case and positioning of TLC downstream of the obstruction may be insightful for better understanding of an underwater robot holding station with trout-Like Configuration in the presence of an obstruction. I believe this work fits well within the scope of Fluids and may merit publication in this journal after addressing following comments/suggestions:
1) Sec.2 Materials and Methods: The definition of the stress tensor $\tau_{ij}$ needs to be given.
2) Sec.2 - Line 122: Though not mandatory, in general I would recommend using consistent unit system throughout the manuscript (either SI or CGM).
This prevents any possible confusion in reproducing results.
3) Sec.2 - Line 126: ``low Reynolds numbers'' could be misleading as it could be all relative. I would suggest providing a range or a clearer identification of a flow regime.
4) Sec.2: In subsection 2.1, a brief description of handling fluid-structure interaction (FSI) in ANSYS needs to be added. Is it using body-fitted arbitrary Lagrangian Eulerian approach? What are the FSI boundary conditions?
5) Sec.2: There seems to be an issue with figure numbering, as Fig.1 has been repeated twice. Recommend either putting the two in one Fig (two panels) or separate numbering.
6) About the UDF, DEFINE_GRID_MOTION: Unless the code has been added as a supplementary material, mentioning just the name of the function may be unnecessary.
7) Again in numbering the figures, right now Fig. 8 seems to be appearing right after Fig. 4.
8) In the pressure plot of the flow around the obstruction (currently Fig.4), have authors observed any vortex shedding behavior? It is hard to tell from the current pressure contour but has the simulation run long enough to observe such possible behavior?
9) Sec.5: Conclusion: line 335-336: This claim is not justified. The present work does not introduce a new FSI approach and therefore saying that ``it will be helpful in the development of numerical models for the hydrodynamic analysis of the bio-inspired UUV systems'' is not accurate, to say the least. The focus here can be instead, on understanding the dynamics and not the computational approach.

Round 2
Reviewer 1 Report
It would be good to have some comparison with experimental observations or measurements!
Author Response
May 27, 2021
Reviewer#1 Comments – Round 2
Comment 1 - It would be good to have some comparison with experimental observations or measurements!
Our response: The following statement is placed in the “Conclusions” section to address the reviewer’s comment.
“The first phase of this research effort was focused on a natural swimmer taking advantage of an obstruction in the flow path. However, the numerical model employed in this study can be enhanced with data obtained from additional experimental work using mechanical fish. Therefore, our future work will include experimental efforts using mechanical fish configurations capable of mimicking various natural swimmers’ oscillatory motions in the laboratory setting.”
